# The Association between Wearing a Mask and COVID-19

**DOI:** 10.3390/ijerph18179131

**Published:** 2021-08-30

**Authors:** Mana Sugimura, Odgerel Chimed-Ochir, Yui Yumiya, Hiroki Ohge, Nobuaki Shime, Takemasa Sakaguchi, Junko Tanaka, Toshiro Takafuta, Michi Mimori, Masao Kuwabara, Toshimasa Asahara, Eisaku Kishita, Tatsuhiko Kubo

**Affiliations:** 1Department of Public Health and Health Policy, Graduate School of Biomedical and Health Sciences, Hiroshima University, Hiroshima 734-8553, Japan; b186361@hiroshima-u.ac.jp (M.S.); odgerel@hiroshima-u.ac.jp (O.C.-O.); yumiya@hiroshima-u.ac.jp (Y.Y.); 2Department of Infectious Diseases, Hiroshima University Hospital, Hiroshima 734-8551, Japan; ohge@hiroshima-u.ac.jp; 3Department of Emergency and Critical Care Medicine, Graduate School of Biomedical and Health Sciences, Hiroshima University, Hiroshima 734-8551, Japan; nshime@hiroshima-u.ac.jp; 4Department of Virology, Graduate School of Biomedical and Health Sciences, Hiroshima University, Hiroshima 734-8553, Japan; tsaka@hiroshima-u.ac.jp; 5Department of Epidemiology, Infectious Disease Control and Prevention, Graduate School of Biomedical and Health Sciences, Hiroshima University, Hiroshima 734-8553, Japan; jun-tanaka@hiroshima-u.ac.jp; 6Hiroshima City Funairi Citizens Hospital, Hiroshima 730-0844, Japan; toshiro.takafuta@gmail.com; 7Hiroshima City Health and Welfare Bureau, Hiroshima 730-8586, Japan; mimori-m@city.hiroshima.lg.jp; 8Hiroshima Prefectural Center for Disease Control and Prevention, Hiroshima 730-8511, Japan; m-kuwabaras3471@pref.hiroshima.lg.jp; 9Hiroshima Prefectural Health and Welfare Bureau, Hiroshima 730-8511, Japan; t-asahara73565@pref.hiroshima.lg.jp (T.A.); e-kishita74732@pref.hiroshima.lg.jp (E.K.)

**Keywords:** COVID-19, J-SPEED, mask, prevention, surveillance

## Abstract

With the widespread and increasing number of cases of Coronavirus Disease (2019) globally, countries have been taking preventive measures against this pandemic. However, there is no universal agreement across cultures on whether wearing face masks are an effective physical intervention against disease transmission. We investigated the relationship between mask wearing and COVID-19 among close contacts of COVID-19 patients in the Hiroshima Prefecture, Japan. In the Hiroshima Prefecture, a COVID-19 form adapted from the reporting form, “Japanese Surveillance in Post-Extreme Emergencies and Disasters”, was developed to collect data from COVID-19 patients’ close contacts under active epidemiological surveillance at Public Health Centers. The relative risk of COVID-19 for mask users versus non-mask users was calculated. A total of 820 interviewees were included in the analysis and 53.3% of them responded that they wore masks. Non-mask users were infected at a rate of 16.4%, while mask users were infected at a rate of 7.1%. Those who wore masks were infected at a rate of 0.4 times that of those who did not wear masks. (RR = 0.4, 95%CI = 0.3–0.6; Adjusted RR = 0.6, 95%CI = 0.3–0.9). These findings implied that COVID-19 could be avoided to a certain degree by wearing a mask.

## 1. Introduction

With the widespread and increasing number of cases of coronavirus disease (2019) (COVID-19) globally, countries have been taking preventive measures, such as wearing masks, to combat the pandemic. However, there is no universal agreement across cultures on whether wearing face masks is an effective physical intervention against disease transmission.

Wearing a mask has become commonplace as a method of infection prevention in Asian countries [1], whereas some health authorities discouraged universal mask use, claiming that masks had no effect in preventing COVID-19, making mask wearers in Western countries feel uncomfortable [2]. Authorities and health care experts all over the world have agreed on the use of masks in health care facilities; however, whether wearing masks is effective in community settings remains debatable. Several attempts have been made to evaluate the effects of mask wearing on COVID-19 prevention. Numerous studies [3,4] support mask use as a preventive measure, while others found no significant link between mask use and COVID-19 [5]. The target study group, study designs, evaluation methods, and endpoints of these studies differed. The meaning of “wearing a mask” varied greatly, particularly in methodological contexts. For example, Mitze et al. compared people who lived in areas where there was a policy requiring them to wear masks to those who lived in areas where there was no such policy [4]. In another study, those who worked were labeled as part of the “mask-on” community, while those who engaged in recreational activities were labeled as part of the “mask-off” community [6]. Murakami et al. used video footage and artificial intelligence [7] to estimate the proportion of people wearing masks. In these studies, statistical modeling was used to estimate mask effects. As such, the number of observational studies investigating the effect of wearing masks to prevent COVID-19 is still limited.

In Japan, Public Health Centers have been conducting active epidemiological surveillance across the country in accordance with the Infectious Disease Control Law, tracking patients’ close contacts and performing Polymerase Chain Reaction (PCR) tests [8]. This surveillance enabled us to collect information on the contact, mask-wearing behavior and PCR tests of those who had contact with infected patients. In this study we investigated the relationship between mask wearing and COVID-19 infection by analyzing data collected through the active epidemiological surveillance implemented in Japanese Public Health Centers.

## 2. Materials and Methods

### 2.1. Data Collection

Following the Great East Japan Earthquake of 2011, a standard medical reporting form, known as J-SPEED (Japanese Surveillance in Post-Extreme Emergencies and Disasters) was developed with the concept of simplifying and standardizing medical reporting by front-line healthcare workers in order to collect almost real-time data [9]. In the Hiroshima Prefecture, Japan, a J-SPEED-style COVID-19 reporting form was developed to collect data from contacts of COVID-19 patients under the active epidemiological surveillance at Public Health Centers. 

The contact information of persons who were diagnosed with COVID-19 (referred to as patients) at the PCR center and all medical facilities in Hiroshima was provided to the Public Health Center. Staff at the Public Health Center, including public health nurses, contacted these patients and identified their close contacts as soon as they received patient information. All close contacts were interviewed and provided with a PCR test by the Public Health Center. Based on the interview record, a J-SPEED-style COVID-19 reporting form was completed.

The J-SPEED-style COVID-19 reporting form included 60 items such as demographic information, contact types, mask-wearing status at the time of contact with patients, and their PCR test result. Those who has a positive PCR test were considered to be “infected”.

We extracted items from the total of 60 that were related to the present study’s objective. For example, we only included information on types of contact, mask-wearing status, site of contact, relationship with patient, and PCR test results, among other things. We did remove items that were connected to clinical symptoms. Following that, we combined some similar items into a single item and excluded items with fewer than 100 cases.

As such, staff at the Public Health Center interviewed a total of 1434 close contacts between 6 March and 31 May 2020 in Hiroshima. Then, we included 820 people out of 1434 interviewees in the analysis who provided answers regarding mask use and had a PCR test.

As defined by the Infectious Disease Surveillance Center, National Institute of Infectious Diseases of Japan, close contact means: (i) persons who are living together who have been in prolonged contact (including inside a car or airplane, etc.) with a person who is suspected of being infected with the novel coronavirus; (ii) persons who have been consulting, nursing, or caring for patients suspected of being infected with the novel coronavirus, without appropriate protection against infection; (iii) and persons who are highly likely to have been in direct contact with contaminants, such as respiratory secretions or the body fluid of a person suspected of being infected with the novel coronavirus; iv) persons who have been in contact with a “patient (confirmed case)” without the necessary preventive measures against infection, at a distance that allows touching by hand or a face-to-face conversation (roughly 2 m). The infectivity of the patient was comprehensively judged based on the patient’s symptoms, etc. [8]. In this context, a patient is defined as someone who met interviewees before realizing that they were infected but was later diagnosed as having COVID-19, via a PCR test.

### 2.2. Data Analysis

We calculated the relative risk of infection for mask users versus non-mask users in the current study. The relative risk was adjusted with gender the and type of contact. Other variables such as age, contact source, and cumulative contact time were intermediate variables on a causal path between exposure (masking) and outcome (positive PCR test). Thus, these intermediate variables were not adjusted in the model to avoid overadjustment bias [10]. Relative risk is also calculated for specific items such as sex, age, type of contact, type of incident, place of contact, relationship with patients and cumulative contact time with the patient. The risk ratio was calculated using Poisson regression, and because the outcome variable is binary (PCR tests were positive or negative), the exponentiated coefficients were regarded as risk ratios rather than incidence-rate ratios [11,12].

The missing values were excluded from the analysis. Microsoft Excel and STATA v15.1 (STATACorp, College Station, Texas, USA) were used for analysis.

## 3. Results

Figure 1 compares the use of masks and the risk of infection between mask users and non-mask users. After removing missing data, 820 out of 1,434 interviewees were included in the analysis. Non-mask users were infected at a rate of 16.4%, while mask users were infected at a rate of 7.1%. Those who wore masks were infected at a rate of 0.4 times compared to those who did not wear masks. (RR = 0.4, 95%CI = 0.3–0.6; Adjusted RR = 0.6, 95%CI = 0.3–0.9).

In men (RR = 0.4, 95%CI = 0.3–0.7), cluster cases (RR = 0.6, 95%CI = 0.2–0.6), welfare services (RR = 0.6, 95%CI = 0.4–0.9), and work relationships (RR = 0.3, 95%CI = 0.2–0.5), there was a significant relationship between mask wearing and being infected with COVID-19, but no significant relationship was found in other items. In welfare services, non-mask wearers had a positive rate of 46.8%, while mask wearers have a positive rate of 27.7%, which is higher than for other items.

Among the total interviewees, 53.3% responded that they were wearing masks. In terms of sex, 50.1% of men and 57.9% of women wore masks. Sixty three percent of people aged 20 to 59 wore masks while only 37.5% of those over 60 did not wear masks. 

Approximately half of those who had close (54.3%) and cluster (53.5%) contacts with patients wore a mask. Only 15.6% of people wore masks at home, while 89.7% of those contacted with patients in medical facilities wore masks. When they were with partners and family, very few people (4.8–12.7%) wore masks. More than two thirds of those who met with patients for less than 15 min wore masks, while those who met with patients for more than 15 min did not. Only 10% of those polled stated that they wore masks while eating with patients.

## 4. Discussion

The current study found that mask wearers mask wearers were 60% (0.4 times) less likely to be infected with COVID-19 than non-mask wearers. 

Many studies have shown that wearing a mask during a pandemic of a respiratory infectious disease had a significant effect [2,3,4,13]; however, differences in the target group, study design, and mask-wearing definition make direct comparisons between studies difficult. In the current study, we discovered a significant association between wearing a mask and not being infected with COVID-19 based on an analysis of data collected from those who had contact with patients and were interviewed at public health centers in Japan.

We investigated this association in specific contexts using stratified analysis by gender, age, contact type, and contact location. Men were found to have a significant association between wearing a mask and not being infected with COVID-19, whereas the association was not as significant among women. However, we did not find any studies that reported on mask effectiveness differences between men and women. This non-significant difference in COVID-19 infection between mask users and non-mask users might be explained by the fact that people did not wear masks properly (loose fitting, not covering both the nose and mouth, touching the mask while wearing it, re-wearing used masks, etc.), despite their responses. Furthermore, we were unable to determine the type of masks used by interviewees when they were in contact with COVID-19 patients. As a result, more males may have used cloth masks, which have less effective filtering efficacy in general [14]. In addition, men were more vulnerable to COVID-19 infection [15] due to several factors such as genetics [16], immunology [17] and behavioral factors [18]. As a consequence, men may have demonstrated a strong association between wearing a mask and not becoming infected with COVID-19.

There was no significant relationship between wearing a mask and not being infected with COVID-19 among those who interacted with patients while having a meal together. It could be explained by insufficient mask use, as patients and interviewees undoubtedly removed masks while eating, despite wearing masks at other times. As a result, in this context, the difference between mask wearer and non-wearer is highly likely to be biased.

There was no significant association between mask use and COVID-19 infection among those who had been in contact with patients at “shops and restaurants,” “home,” or for “more than 15 min.” These were the most common “close-contact settings,” which was one of the “3C” situations (the other two being closed spaces and crowded places) [19]. Previous research found that people in households were less likely to use masks properly in 3C situations than in other situations [20].

As a result, as previously stated, the absence of a significant difference in COVID-19 infection between mask users and non-users might be explained by the fact that people did not adequately wear masks, notwithstanding their responses.

In welfare services, non-mask wearers had a positive rate of 46.8%, while mask wearers have a positive rate of 27.7%, which was higher than for other items. The high positive rate in welfare settings could be explained by the fact that, in welfare facilities, active epidemiological surveillance and PCR tests were mostly performed in the event of a COVID-19 cluster. According to our findings, 61% of infected non-mask wearers in welfare services were over the age of 60. This might be explained by the possibility that older people have difficulties in remembering safeguard procedures, such as wearing masks, or in understanding the public health information issued to them [21], so the positive rate is higher than for other items.

The proportion of people who wore masks varied greatly depending on the situation. Women, for example, appeared to wear masks more often than men, and some studies found that women were more active in preventing infection than men [22,23].

People aged 20 to 59 were the most likely to wear masks, while those aged 60 and up were the least likely. A study conducted in the United States discovered that older people were more likely to wear masks than younger people [24], whereas another study conducted in Taiwan discovered that older people were less likely to wear masks [25]. We considered the possibility that older people did not receive adequate information about mask use, which may have contributed to the lower rate of mask usage among the elderly.

We also discovered that people did not wear masks when they were at home, with their family, partners, or while eating and drinking. According to Al Naam et al., the proportion of people who wore masks at social gatherings was lower than in public places and workplaces [26]. In other words, people did not wear masks when they were around very familiar people, such as family, partners, and close friends. Wearing masks while being with family and partners at home was impractical; therefore, people may be advised to wear masks while they are out to avoid bringing an infection into the home.

The use of active epidemiological surveillance data for analysis was the current study’s main strength. However, this study had several challenges and limitations that are worth mentioning. Firstly, mask wearers may have had a higher level of health literacy, and thus may have used masks in addition to other preventive measures such as hand washing and disinfection. As a result, the current outcome may have been a mixed effect of mask use and other preventive measures and may have overestimated prevention through mask use.

Secondly, we only asked whether or not respondents wore masks when they had contact with patients, but we did not investigate if they wore masks correctly, if they properly changed mask properly, or the types of masks they wore. Thirdly, because data were collected retrospectively from interviewees, there may have been some recall bias regarding their use of masks during contact with patients. Fourthly, there were many missing values in the data collection because the content obtained from the interview survey conducted by the public health nurse was posted to the checkbox by another person in charge. However, comparability was guaranteed because the data were obtained by performing an internal comparison among the obtained data. Furthermore, when the risk ratio was analyzed in context-specific items, we encountered a small sample size problem for some of them. Therefore, more information was required to further analyze the context-specific situation.

## 5. Conclusions

The analysis of data collected through an active epidemiological surveillance at public health centers in the Hiroshima Prefecture, Japan, revealed that mask wearers were 60% less likely to be infected with COVID-19 than non-mask wearers.

## Figures and Tables

**Figure 1 ijerph-18-09131-f001:**
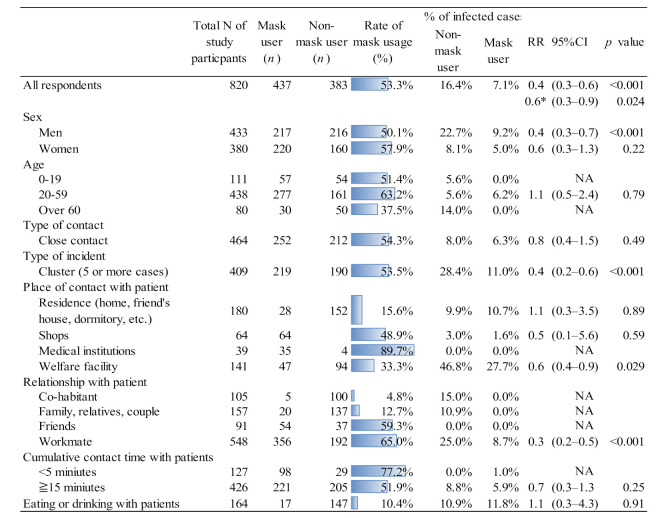
Risk of COVID-19 infection for mask users and non-mask users, and rate of mask use at public health centers in Japan. * Adjusted with sex and contact type.

## Data Availability

Data was obtained from Hiroshima Prefecture and are available with the permission of Hiroshima Prefecture.

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
