# Peer review of "The Association between Wearing a Mask and COVID-19"

_ijerph, 2021, doi:10.3390/ijerph18179131_

Round 1

Reviewer 1 Report

You said:

"Men were found to have a significant association between wearing a mask and not being infected with COVID-19, whereas the association  was not significant among women. However, we did not find any studies that reported on mask effectiveness differences between men and women".

Although you did not find any studies that reported on mask effectiveness differences between men and women. 
What could be the cause? What could you propose as a theory? 

"As a result, the non-significant difference in COVID-19 infection between mask users and non-users may be explained by the fact that people did not wear masks properly despite responding as they did". 

It is logical ... It consists of using the mask but in the proper way, in good condition, etc. 

Author Response

Author’s note:

Reviewer #1:

Comment 1:   You said: "Men were found to have a significant association between wearing a mask and not being infected with COVID-19, whereas the association was not significant among women. However, we did not find any studies that reported on mask effectiveness differences between men and women".

Although you did not find any studies that reported on mask effectiveness differences between men and women.  What could be the cause? What could you propose as a theory?

"As a result, the non-significant difference in COVID-19 infection between mask users and non-users may be explained by the fact that people did not wear masks properly despite responding as they did".

It is logical ... It consists of using the mask but in the proper way, in good condition, etc.

Our response:  Thank you very much for your valuable comment. We incorporated your comment into the revised version (Line 174-184).

Reviewer 2 Report

In the Brief Report  titled “ Association between Wearing Mask and COVID-19” the authors nvestigated the relationship between mask wearing and COVID-19 among close contacts of COVID-19 patients in Hiroshima Prefecture, Japan. In particular they calculate relative risk of COVID-19 for mask users versus non-mask users.

I thing that this work was very interesting but certain points need to be checked before suggesting publication. I suggest a major revision.

I think that it is not sufficiently clear the type of exposition of the partecipants. It is necessary to describe better

It is necessary to precise the type of masks used. It is also important to know if the masks were changed every day or not and if they were used in the right way

Pollution affects covid severity. The role of Hiroshima pollution and covid could be discussed. For this aim I suggest he authors to read and quote the following papers:

10.3390/ijerph18136846

10.1007/s11356-021-14579-x.

Finally, I think that all possible variables that may influence the results presented in this paper could be taken into account to define more significant and reliable values. The authors must be sure of the values obtained. A revision of the results, if there are variables not considered is welcome.

Author Response

19 August 2021

MS ID#: ijerph-1337737

MS TITLE:  Association between Wearing Mask and COVID-19

Point-by-Point Response Letter

Thank you for the valuable comments in reviewing our manuscript. The following is our point-by-point responses which were reflected in the revised version of the manuscript. The manuscript has been revised, with changes highlighted in red font.

Reviewer #2:

Comment 1. I think that it is not sufficiently clear the type of exposition of the participants. It is necessary to describe better

Our response:  Following the suggestion by the reviewer, we incorporated additional information on participant descriptions. (Line 85-91)

Comment 2. It is necessary to precise the type of masks used. It is also important to know if the masks were changed every day or not and if they were used in the right way

Our response:  One drawback of the current study is that we were unable to establish how correctly used masks or what sorts of masks they used in the form used at Public Health Canter. As a result, we mentioned it as a limitation in the manuscript. (Line 233-235)

Comment 3. Pollution affects covid severity. The role of Hiroshima pollution and covid could be discussed. For this aim I suggest he authors to read and quote the following papers:

10.3390/ijerph18136846

10.1007/s11356-021-14579-x.

Our response:  Thank you very much for your valuable comment. We found that these two publications provided a good summary of the influence of air pollution on the severity of COVID-19. Furthermore, the primary objective of these two studies was to synthesize the evidence of a synergistic impact of COVID-19 and air pollution on male fertility. However, during the research period (March 2020), the level of PM2.5 in Hiroshima was 8.4 g/m3, which is four times lower than the standard threshold (35 g/m3). Please refer to the following documents: https://www.pref.hiroshima.lg.jp/uploaded/attachment/416320.pdf As a result, we believe that the impact of air pollution on the severity of COVID-19 in Hiroshima was less likely. Furthermore, we were unable to conclusively establish a link between COVID-19, mask use, and sperm quality in the presence of air pollution and COVID-19.

Comment 3. Finally, I think that all possible variables that may influence the results presented in this paper could be taken into account to define more significant and reliable values. The authors must be sure of the values obtained. A revision of the results, if there are variables not considered is welcome.

Our response: We understood this comment regarding "whether or not we overlooked any essential items  (out of 60 items reported in the reporting form) to include in the analysis."  If this is the case, please see our response below; otherwise, please accept our sincerest apologies for any confusion.

We only extracted items from a total of 60 that we thought were related to the present study's objective. For example, we only included information on types of contact, mask wearing status, site of contact, relationship with patient, and PCR test results, among other things. We did, however, remove items that were connected to clinical symptoms. This clarification was made to the paper.

EOF

Round 2

Reviewer 2 Report

Accept in present form